# Preliminary Research on the Influence of a Pulsed Magnetic Field on the Cationic Profile of Sunflower, Cress, and Radish Sprouts and on Their Germination Rate

Grzegorz Zaguła *, Bogdan Saletnik, Marcin Bajcar, Aneta Saletnik and Czesław Puchalski

Department of Bioenergetics, Food Analysis and Microbiology, Institute of Food Technology and Nutrition, College of Natural Sciences, University of Rzeszów, Ćwiklińskiej 2D, 35-601 Rzeszów, Poland; bogdan.saletnik@urz.pl (B.S.); mbajcar@ur.edu.pl (M.B.); a.saletnik@ur.edu.pl (A.S.); cpuchal@ur.edu.pl (C.P.)
* Correspondence: g_zagula@ur.edu.pl

**Abstract:** Magnetic stimulation of seeds before sowing can have a significant impact on the speed of their germination. Sprouts are sought after by consumers for their high nutrient content. The purpose of the study was to investigate the influence of a pulsed magnetic field on the dynamics of seed germination and on the content of ions in sunflower, cress, and radish sprouts. The research material in the experiment was provided by seeds of sunflower (*Helianthus annuus* L.), garden cress (*Lepidium sativum* L.), and garden radish (*Raphanus sativus* L.) intended for sprouting, which were supplied by PNOS Ożarów Mazowiecki. The research methods involved germinating seeds under strictly defined conditions for 14 days. Then, the mineral composition of the previously mineralised sprout material was determined using emission spectrometry on a ICP-OES iCAP Duo 6500 Termo spectrometer. Greater dynamics of germination were noted in the first half of the growth period in seeds stimulated with a pulsed magnetic field with the parameters 100 μT and 100 Hz. However, the application of the magnetic field produced no increase in the capacity of the seeds to germinate. The research showed an increase in the content of macronutrients in sprouts, such as calcium, magnesium, phosphorus, and sulphur. In the case of the field with parameters of 100 μT and 200 Hz, the effect was similar for both the germination percentage and the accumulation of macronutrients. However, in the case of both frequencies of magnetic field applied, the effect on individual plant seed species was different. Pre-sowing stimulation of seeds with a pulsed magnetic field may affect the rate of seed germination and the content of ions in the sprouts; however, these effects vary in individual plant matrices.

**Keywords:** sprouts; stimulation with a pulsed magnetic field; micro and macro components; ICP-OES

## 1. Introduction

Seed germination is a process consisting of three phases, referred to as the imbibition phase, in which there is intensive cellular respiration and water uptake, the catabolic phase, where hydrolysis and metabolic processes again take place, and the third phase, referred to as anabolic, in which further growth and development are observed [1]. In Phase I, rapid water uptake takes place, which initiates the metabolism in which DNA and mitochondrial repair occurs along with protein synthesis using existing mRNA. During Phase II, further water uptake is limited because the water potential of the grain is almost in balance with that of the surrounding environment. This phase is also referred to as the activation or retardation phase. Major metabolic changes take place in this phase, such as the synthesis of hydrolytic enzymes (e.g., α-amylase, endoxylanase, and phytase) and other processes necessary for the development of the embryo. In phase III, there is a second rapid water intake. The rootlets appear, i.e., the so-called visible germination, hereinafter referred to as germination [2,3]. Sprouts are formed from seeds during germination and are an excellent source of protein, vitamins, and minerals, containing nutrients as important to health as

glucosinolates, plus phenolic and selenium components in cruciferous plants or isoflavones in soybeans. Since sprouts are consumed early in the growth phase, their nutrient concentration remains very high. In addition to nutrients, sprouts contain phytochemicals, vitamins, minerals, enzymes, and amino acids, which are of the greatest importance as they are most beneficial to human health [4]. Sprouts have been consumed in China and eastern countries for many years, and, recently, as a result of a changing lifestyle, sprouts are becoming more and more popular among people around the globe due to their nutritional value and health benefits. A large number of epidemiological studies consistently show that the daily consumption of plant-based foods is associated with a reduction in risk factors for chronic diseases such as cardiovascular diseases, diabetes, and obesity. The health-promoting effects of plant-based foods may be related to the presence of several bioactive compounds in the edible parts, such as phenolic compounds, carotenoids, glucosinolates, vitamin C, and tocopherols, which show various biological properties. Numerous studies have proven that sprouting is an inexpensive and effective method of accumulation of bioactive compounds in the seeds of legumes, cereals, vegetables, fruit, flowers, and medicinal plant seeds [5].

A significant problem in the production of sprouts is their growth rate. Innovative and novel methods of accelerating plant germination are currently sought, including for sprouts [6,7]. The known alternative methods of accelerating the germination process include, inter alia, laser stimulation [8–10], the use of a permanent magnetic field [11,12], and seeds exposed to a static magnetic field [13,14]. Now, a pulsating magnetic field [15], in which stimulating effects acting at least in two planes in space give an opportunity to significantly accelerate the impact of this alternative method on the speed with which a valuable product with increased nutritional potential is obtained, is now also worthy of attention. This is because plant organisms, including seeds, can be treated as specific antennae that receive electrical and magnetic signals from the external environment [16]. This phenomenon is stronger in the case of resonance, and, often, the energy causes biological effects that are many times stronger [17]. The research carried out so far has shown the beneficial effect of the magnetic field on seeds and the plants grown therein [18]. The results obtained are measured by the effect of accelerated germination, the beneficial effect on growth, and the chemical composition of the grown plants. However, the results show that the changes of these parameters depend on many physical factors, such as the exposure dose, the type of magnetic field, the design of the device, or the plant species and cultivar. The issue of the influence of the magnetic field on the development and yield of plants has been largely documented by scientific results; however, a comprehensive explanation of this phenomenon requires further research.

A pulsating magnetic field is a new and very contemporary topic. Authors dealing with the issue of the impact of a PMF direct their research towards, inter alia, stimulation to improve the healing of bone fractures [19–21] and the impact of such a pulsating field at the cellular level [22–26].

The manuscript presents the results of research on the influence of a pulsed magnetic field with induction of 100 µT and two non-ionising frequencies 100 Hz and 200 Hz on the germination potential and mineral composition of three plant species intended for sprouts, i.e., sunflower (*Helianthus annuus* L.), garden cress (*Lepidium sativum* L.), and garden radish (*Raphanus sativus* L.).

## 2. Materials and Methods

The research material in the experiment was provided by seeds of sunflower (*Helianthus annuus* L.), garden cress (*Lepidium sativum* L.), and garden radish (*Raphanus sativus* L.) intended for sprouting and supplied by PNOS Ożarów Mazowiecki.

The study material consisted of 450 seeds for each species, divided into 3 groups. The first group was stimulated with a pulsed magnetic field with the parameters of 100 µT and 100 Hz, the second group was stimulated with a pulsed magnetic field with the parameters

of 100 µT and 200 Hz. The third group consisted of seeds intended for the control sample, which had not been subjected to magnetic stimulation.

### 2.1. Magnetic Stimulation Site

The seeds were subjected to pre-sowing magnetic stimulation with the use of a set of coils generating alternating magnetic fields pulsed in the XY plane. In this system, it is possible to set the field rotation frequency in the range of 20–100 Hz, the field generation time from 1 s to 100 h, and the field magnetic induction value in the range of 0–500 µT. All the aforementioned operating parameters of the device are determined using a PC and an application in the LabVIEW environment. The seeds were subjected to magnetic stimulation with the induction and frequency parameters of 100 µT and 100 Hz, as well as 100 µT and 200 Hz, and the exposure time to the magnetic field was 30 min (Figures 1 and 2).

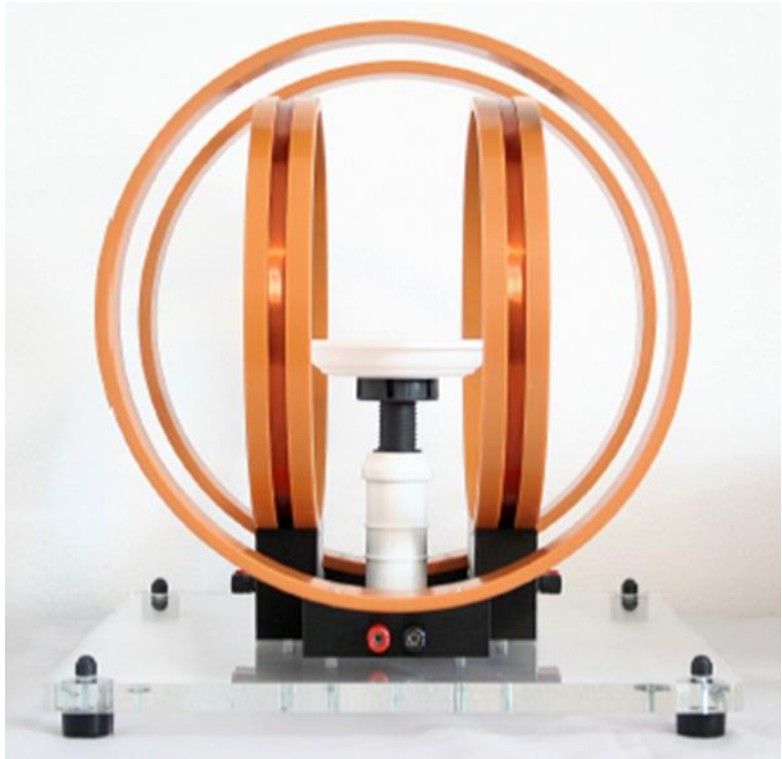

**Figure 1.** A set of coils for generating alternating magnetic fields (own study).

The system used consists of the following elements:

- A 2-channel function generator, DDS-JDS6600, which generates 2 sinusoidal voltage waveforms for channel X(CH1) and Y(CH2) with fixed amplitudes, and the same frequency for both channels.
- The MONACOR SA-200 voltage amplifier, to which the signals from the generator are transferred.
- A signal splitter/current intensity sensor (BOX)—the amplified signal from the amplifier passes to the signal splitter BOX, from where it is transferred to the individual windings of 2 pairs of Helmholtz coils. For channels X and Y, the current intensity is measured by measuring the voltage drop across precision resistors installed in the distributor. Subsequently, the voltage signal from the resistors is transferred from the splitter to the myDAQ digital oscilloscope.
- A group of Helmholtz coils—the current flowing through the coils creates a variable pulsed magnetic field Bxy in the XY plane within them. The X and Y coils consist of two sections, X1 and X2, and Y1 and Y2, attached to the base. These sections, in turn, are connected in series in the BOX distributor so that the current flowing through each

pair of coils has the same value. In the working area, there is a table on which the sample is placed.

- A NI myDAQ device used as a digital oscilloscope, recording voltage waveforms from measuring resistors
- Control software that determines the parameters for the current flowing through the coils and the magnetic field generated, and also controls the work of the JDS6600 generator by determining the frequency and phase shift of the signals and directs the amplitude of the waveforms in real time.

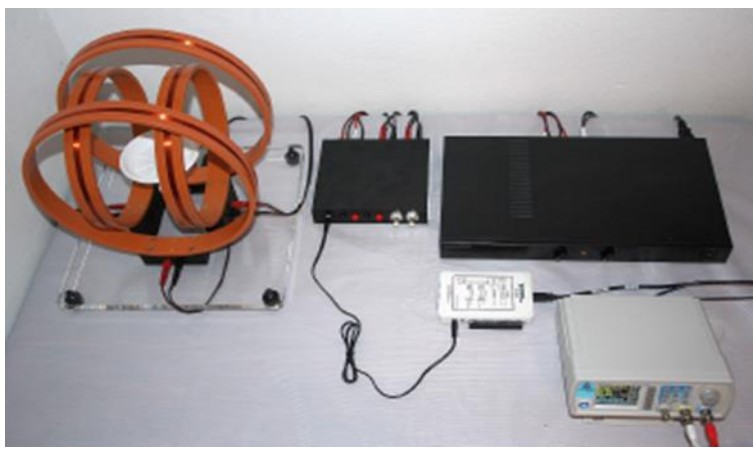

**Figure 2.** View of the overall set-up (own study).

### 2.2. The Dynamics of Seed Germination

Each of the 3 groups of seeds was divided into three samples, with 50 seeds in each, sown on plates. Initially, 100 mL of demineralised water was added to each batch of seeds. The seeds were hydrated for 4 h at 20 °C and then transferred to airtight containers and incubated at 20 °C for 2 days, with daily airing. Incubation was carried out at 20 °C on moist filter paper in Petri dishes and the number of germinated seeds was counted daily. Seeds, the germinal roots of which had pierced the seed coat and had a length of at least 1 mm, were considered to be germinated.

### 2.3. Assessment of the Mineral Composition

After 14 days, all sprouts were cut in order to determine the content of micro and macro elements therein, using the researcher's own procedure.

The samples of sprouts were mineralised under high pressure in super pure 65% HNO3. Samples of 0.5 g were weighed and placed in Teflon vessels, which were then filled with 8 mL of nitric acid and sealed tightly. For each group of nine samples, the rotor of a digestion system was also filled with a blank sample. The samples were digested at an algorithm of temperature increasing as specified for biological samples, never exceeding 200 °C. This procedure was carried out in an Ethos One microwave digestion system from Milestone. The vessels were opened after the mineralisation process had been completed and the samples with acid were brought to room temperature. Afterwards, they were replenished with water to a volume of 50 mL. The detection threshold obtained for each element was not lower than 0.01 mg kg$^{-1}$ (with an assumed detection capacity of the measuring apparatus at a level exceeding 1 ppb). The measurements were performed on a Thermo iCAP Dual 6500 ICP-OES spectrometer with horizontal plasma and with a capacity of detection being determined both along and across the plasma flame (Radial and Axial). Before measuring each batch, the equipment was calibrated with the use of certified Merck models, with concentrations of 10,000 ppm for Ca, Fe, K, Mg, and P, and 1000 ppm for Al, Ba, Cd, Cu, Na, and Pb. The measurement result for each element was adjusted to account for the measurement of elements in the blank sample.

In each case, a 3-point calibration curve was used for each element, with optical correction applying the method of internal models, in the form of ytterbium and ytterbium ions, at concentrations of 2 mg L$^{-1}$ and 5 mg L$^{-1}$, respectively. The analytical methods were validated using two independent tests. Certified Reference Material (NIST—1515) was used and the recovery obtained for specific elements is shown in Table 1. The method of adding a model with a known concentration was applied in order to identify the relevant measurement lines and to avoid possible interferences (Table 1).

**Table 1.** The lengths of measurement lines and the recovery obtained for the specific elements examined.

| Element | Slotted Line | Recovery According to CRM | Recovery According to Method of Standard Addition |
|---|---|---|---|
| | **nm** | **%** | **%** |
| Al | 167.079 | 98 | 99 |
| Ca | 317.933 | 103 | 98 |
| Cd | 228.802 | 99 | 97 |
| Fe | 259.940 | 99 | 97 |
| K | 766.490 | 101 | 97 |
| Mg | 279.533 | 99 | 98 |
| Mn | 257.610 | 98 | 97 |
| Mo | 203.844 | 101 | 99 |
| Na | 588.995 | 98 | 100 |
| Ni | 231.604 | 99 | 100 |
| P | 177.495 | 100 | 101 |
| Pb | 220.353 | 101 | 98 |
| S | 180.731 | 103 | 99 |
| Se | 196.090 | 101 | 98 |
| Zn | 213.856 | 102 | 101 |

*2.4. Statistical Analysis*

All parts of the experiment were independently repeated three times. The results obtained were subjected to statistical analyses using Statistica ver. 10.0. The results were statistically analysed with multiway ANOVA and the differences between the means were assessed using the Tuckey test.

**3. Results and Discussion**

The first parameter analysed throughout the growth period of sunflower, cress, and radish seed sprouts is the germination parameter. The results obtained for radish sprouts clearly show the tendency to accelerate the seed germination process during the first period of germination. After this period has elapsed, the number of germinated seeds in the control samples is equal to the number in the samples stimulated by two selected exposure doses of pulsating magnetic fields (Figure 3). The improvement of the germination process has been of interest to many authors. The factor improving this parameter was described by Menegatti et al. [27]. They indicated that the exposure of passion fruit seeds to a sublime magnetic field in an isolated manner stimulates seed germination, emergence, and vigour. In another text of the authors [28], they demonstrated that chickpea seeds (*Cicer arietinum* L.) subjected to magnetic treatment showed an improvement in seed germination efficiency in terms of the speed and length of the sprouts and the dry matter content of the seedling, and that the reaction varied depending on the intensity of the field and exposure time. The same positive effect on the rate of seed germination and vigour index was found in cucumber seeds (Cucumis sativus L. Var. Barsati) [29], lettuce (*Lactuca sativa* L.) [30],

maize seeds (Zea mays L. Var. HQPM-1) [31], tomato seeds (*Solanum lycopersicum* L. Var. MST/32) [32], and radish seeds [33,34].

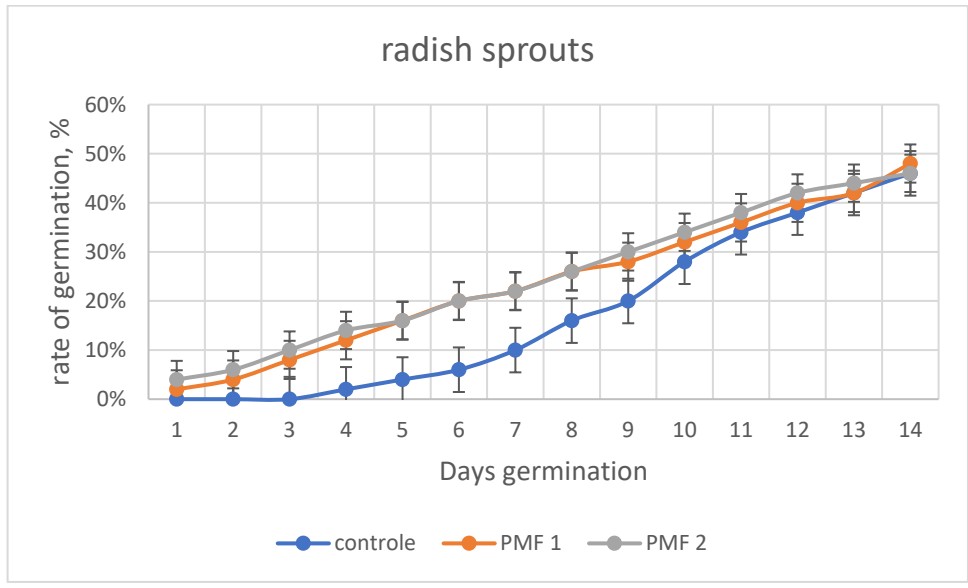

**Figure 3.** Radish sprout germination dynamics for the control sample and for magnetic field 1 (PMF 1–100 µT and 100 Hz) and magnetic field 2 (PMF 2–100 µT and 200 Hz), mean standard error α = 0.05.

Similar results of improvement in germination capacity are seen in trials of growth enhancement effects caused by plasma irradiation of atmospheric air dielectric discharge and thermal shock with radish sprout seeds (*Raphanus sativus* L.). Interactions between radicals and seeds over a short time period of 3 min lead to an enhancement of the growth of radish sprouts over the long period of 7 days, and the maximum average length is 3.7 times larger than in the case of the control. The growth enhancement effects gradually weaken over time, therefore the ratio of mean duration of plasma irradiation to control irradiation decreases from 3.7 on the first day to 1.3 within seven days. The average length of the thermal shock of 60 °C for 10 min and 100 °C for 3 min is longer than in the case of the control, and the maximum average length is 1.3 times longer than in the case of the control. Thermal shock contributes little to the increase due to plasma irradiation because the maximum temperature due to plasma irradiation is less than 60 °C [35].

The results obtained for sunflower sprouts (Figure 4) clearly show the tendency to accelerate the seed germination process for almost the entire germination period up to day 12. After this point, the control samples have the same number of germinated seeds as the samples stimulated by the two selected exposure doses of pulsating magnetic fields. The study [36] shows the results obtained in terms of the effect of treating sunflower seeds subjected to accelerated ageing and cold test on seed vigour. The seeds were treated with distilled water, a solution of potassium nitrate (0.2%), and a solution of gibberellic acid (0.04%). The following parameters were examined: germination energy, germination, the share of abnormal seedlings, length of roots, and shoots of normal seedlings. Accelerated ageing for three and five days resulted in a statistically significant reduction in germination and germination energy; this adversely affected the length of the roots and shoots, and increased the share of abnormal seedlings. Treating the seeds with all three solutions mitigated the adverse effect of three-day accelerated ageing on germination energy. Moreover, treatment of seeds with gibberellic acid prior to the three-day accelerated ageing procedure had a positive effect on seed germination and neutralised the negative effect of accelerated ageing on the number of abnormal seedlings and the length of shoots and roots of normal seedlings. The cold test (at 5 °C for seven days) had a negative effect on the germination energy and root length, increased the share of abnormal seedlings, and did not affect seed germination. Treating seeds with distilled water prior to the cold test

completely neutralised the negative influence of low temperatures on germination energy. Eventually, the treatment of the seeds with all three solutions completely neutralised the adverse effect of the cold test on root length.

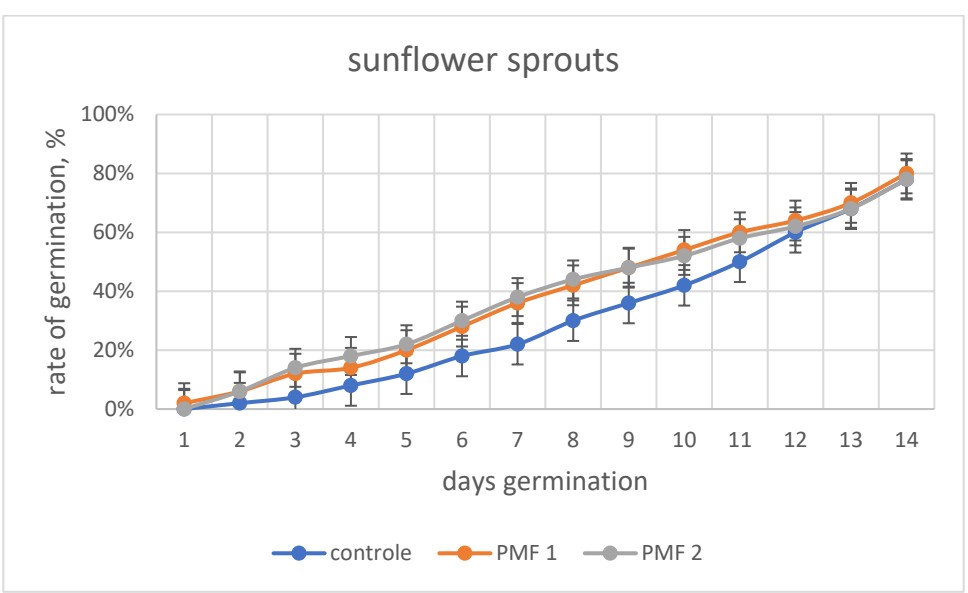

**Figure 4.** Sunflower sprout germination dynamics for the control sample and for magnetic field 1 (PMF 1–100 µT and 100 Hz) and magnetic field 2 (PMF 2–100 µT and 200 Hz), mean standard error $\alpha = 0.05$.

The results obtained for cress sprouts (Figure 5) show a tendency to accelerate the seed germination process by day seven. After this time, the control samples have the same number of germinated seeds with the samples stimulated by the two selected exposure doses of pulsating magnetic fields. Similar results for improved germination capacity are described by the authors using laser stimulation for cress seeds [37–40].

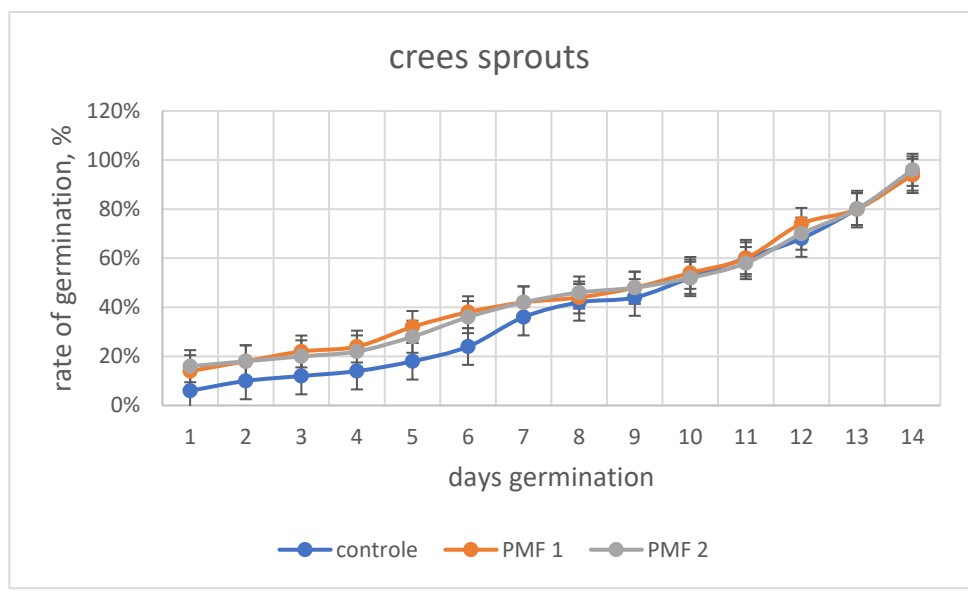

**Figure 5.** Cress sprout germination dynamics for the control sample and for magnetic field 1 (PMF 1–100 µT and 100 Hz) and magnetic field 2 (PMF 2–100 µT and 200 Hz), mean standard error $\alpha = 0.05$.

The use of an increased frequency of magnetic field seems to be irrelevant for the acceleration of the germination processes. This is because the dynamics of germination

improvement are maintained for both 100 Hz and 200 Hz. The determining factor here, therefore, is the magnetic field induction of 100 µT.

Another parameter examined was the total content of micro and macro components in seed sprouts (Table 2). As a reservoir of valuable bio-nutrients, sprouts are especially valuable due to the content of basic micro and macro components. In recent years, sprouts and micro-herbs have been of great interest in many disciplines [41–44].

**Table 2.** Changes in the content of basic micro and macro components and heavy metals under the influence of applied pulsed magnetic fields for cress sprouts.

| Ions | Control | PMF 1 | PMF 2 |
|---|---|---|---|
| | mg/100 g | mg/100 g | mg/100 g |
| Al | 2.07 ± 0.86 [a] | 2.20 ± 1.27 [a] | 2.23 ± 1.28 [a] |
| Ca | 492.51 ± 13.91 [c] | 540.92 ± 5.12 [b] | 699.75 ± 5.65 [a] |
| Cd | <LOQ | <LOQ | <LOQ |
| Fe | 2.50 ± 0.25 [a] | 2.99 ± 0.84 [a] | 2.60 ± 1.21 [a] |
| K | 877.92 ± 52.04 [b] | 981.83 ± 7.91 [a] | 858.08 ± 5.43 [b] |
| Mg | 324.25 ± 25.00 [c] | 352.08 ± 2.69 [b] | 379.91 ± 2.98 [a] |
| Mn | 0.72 ± 0.08 [b] | 1.01 ± 0.04 [a] | 1.05 ± 0.12 [a] |
| Mo | 0.05 ± 0.01 [a] | 0.07 ± 0.01 [a] | 0.10 ± 0.02 [a] |
| Na | 588.75 ± 6.14 [a] | 577.51 ± 4.86 [a] | 560.75 ± 3.07 [a] |
| Ni | <LOQ | <LOQ | <LOQ |
| P | 333.31 ± 14.43 [b] | 373.66 ± 2.25 [a] | 371.00 ± 0.66 [a] |
| Pb | <LOQ | 0.02 ± 0.01 [a] | 0.02 ± 0.01 [a] |
| S | 631.56 ± 14.43 [c] | 704.73 ± 0.66 [b] | 830.90 ± 0.14 [a] |
| Se | <LOQ | 0.06 ± 0.03 [a] | 0.07 ± 0.05 [a] |
| Zn | 5.67 ± 1.15 [b] | 4.91 ± 0.12 [b] | 8.36 ± 0.01 [a] |

Mean values with ±SD and statistical analysis using Tukey's test at the significance level of $\alpha = 0.05$; the results indicated with the same letters do not demonstrate a statistically significant difference (test performed as part of a comparison of the concentrations of individual ions with each other after the application of slowly changing magnetic fields and with a control group); <LOQ—result below the Limit of Quantification.

By analysing the results obtained, it is possible to identify the influence of the pulsed magnetic fields on the ionic composition of the cress sprouts. The highest statistically significant increases in the ion content were recorded for calcium with a magnetic field with a frequency of 200 Hz. A similar trend was observed for magnesium and sulphur. On the other hand, the magnetic field with a frequency of 100 Hz was more favourable and more effective in increasing the content of such ions as potassium and phosphorus. However, it should be clearly stated that the magnetic fields applied have a positive effect on the content of macronutrients in particular. Numerous authors have been involved in the comparison of changes in the content of the basic macro- and micro elements during the growth process [45,46].

By analysing the results obtained, it is possible to identify the influence of the pulsed magnetic fields on the ionic composition of the radish sprouts (Table 3). The highest statistically significant increases in ion content were recorded for calcium with a magnetic field with a frequency of 100 Hz. A similar trend was observed for potassium, magnesium, and sulphur. On the other hand, the magnetic field with a frequency of 200 Hz was more favourable and resulted in an increase in the content of such ions as iron and was less beneficial in relation to sodium from the nutritional point of view. Radish sprouts contain particularly large amounts of these minerals, the amounts of which increased after the application of a magnetic field with a frequency of 100 Hz. These amounts are also confirmed by research [47] and valuable bioactive ingredients as in the following studies [48–50].

**Table 3.** Changes in the content of basic micro and macro components and heavy metals under the influence of applied pulsed magnetic fields for radish sprouts.

| Ions | Control | PMF 1 | PMF 2 |
|---|---|---|---|
| | mg/100 g | mg/100 g | mg/100 g |
| Al | 1.47 ± 0.66 [a] | 2.53 ± 1.12 [a] | 1.59 ± 0.54 [a] |
| Ca | 626.33 ± 6.97 [b] | 802.58 ± 3.26 [a] | 658.50 ± 5.07 [b] |
| Cd | 0.02 ± 0.01 [a] | 0.03 ± 0.01 [a] | 0.01 ± 0.01 [a] |
| Fe | 4.77 ± 1.05 [b] | 7.25 ± 1.24 [a] | 4.41 ± 0.88 [b] |
| K | 659.08 ± 4.93 [c] | 1001.25 ± 2.17 [a] | 846.83 ± 6.08 [b] |
| Mg | 344.58 ± 3.46 [c] | 489.41 ± 1.60 [a] | 390.83 ± 2.74 [b] |
| Mn | 1.61 ± 0.15 [c] | 2.37 ± 0.18 [a] | 2.00 ± 0.08 [b] |
| Mo | 0.05 ± 0.01 [a] | 0.06 ± 0.01 [a] | 0.05 ± 0.01 [a] |
| Na | 315.08 ± 3.59 [c] | 543.91 ± 2.02 [a] | 485.16 ± 2.67 [b] |
| Ni | <LOQ | <LOQ | <LOQ |
| P | 353.35 ± 15.48 [c] | 762.75 ± 1.28 [a] | 661.58 ± 3.02 [b] |
| Pb | <LOQ | <LOQ | <LOQ |
| S | 738.40 ± 5.62 [c] | 1137.48 ± 6.50 [a] | 914.48 ± 5.87 [b] |
| Se | <LOQ | 0.15 ± 0.02 [a] | 0.11 ± 0.01 [b] |
| Zn | 7.03 ± 0.06 [c] | 11.39 ± 0.03 [a] | 8.69 ± 0.04 [b] |

Mean values with ±SD and statistical analysis using Tukey's test at the significance level of $\alpha = 0.05$; the results indicated with the same letters do not demonstrate a statistically significant difference (test performed as part of a comparison of the concentrations of individual ions with each other after the application of slowly changing magnetic fields and with a control group); <LOQ—result below the Limit of Quantification.

By analysing the results obtained, it is possible to identify the influence of the pulsed magnetic fields on the ionic composition of the sunflower sprouts (Table 4). The highest statistically significant increases in the ion content were recorded for calcium with a magnetic field with a frequency of 200 Hz. A similar trend was observed for magnesium, potassium, phosphorus, and sulphur. On the other hand, the magnetic field with a frequency of 100 Hz was more favourable and effective in increasing the content of such ions as sodium and iron.

The dominant trend seems to be a beneficial effect of the use of a pulsed magnetic field on the growth of bound components that act as electrolytes in the human body. These are extremely important ingredients for the body's nutrition. In the clinical setting, the measurement of electrolytes or substances dissolved in the blood is one of the most frequently ordered laboratory tests. The test is known by many names, including electrolyte, "solid", basal metabolic profile (BMP), serum chemistry, CHEM-7, CHEM-10, and sequential multiple analysis-7 (SMA-7). Interpretation of the results provides information on the volume status, acid-base status, and basal renal function. Assessment of any electrolyte imbalance can be difficult and requires an understanding of the pathophysiology of diseases that may cause electrolyte imbalance, the body's counter-regulatory pathways to correct imbalances, and various approaches to increasing the amount of a specific electrolyte through the use of drugs, replacement of a specific electrolyte, or electrolyte removal. Understanding these disorders can improve patient care, be cost-effective, prevent complications from the primary disease, and ultimately reduce mortality and morbidity [51].

The summary of the sum of minerals, including the division into micro–macro elements and heavy metals, including their fluctuations after stimulation with a pulsating magnetic field, turned out to be particularly interesting (Figures 6–8). The division into two groups of sprouts was clearly visible. A magnetic field with a frequency of 200 Hz had a better effect on cress sprouts and their saturation with macronutrients and microelements, while a pulsed field with a frequency of 100 Hz was more favourable for sunflower and

radish sprouts. These results suggest the advisability of research into the use of a pulsed magnetic field and the matching of the magnetic field parameters to the subject of research.

**Table 4.** Changes in the content of basic micro and macro components and heavy metals under the influence of applied pulsed magnetic fields for sunflower sprouts.

| Ions | Control | PMF 1 | PMF 2 |
| --- | --- | --- | --- |
| | mg/100 g | mg/100 g | mg/100 g |
| Al | 0.32 ± 0.06 [a] | 0.28 ± 0.12 [a] | 0.32 ± 0.04 [a] |
| Ca | 429.00 ± 1.88 [c] | 543.85 ± 3.74 [b] | 613.25 ± 2.00 [a] |
| Cd | 0.02 ± 0.01 [a] | 0.02 ± 0.01 [a] | 0.03 ± 0.01 [a] |
| Fe | 4.98 ± 0.25 [a] | 5.66 ± 0.38 [a] | 5.45 ± 0.58 [a] |
| K | 751.12 ± 1.32 [c] | 1380.25 ± 4.76 [a] | 1074.08 ± 6.12 [b] |
| Mg | 560.50 ± 1.80 [b] | 649.33 ± 4.50 [a] | 544.67 ± 2.18 [b] |
| Mn | 1.61 ± 0.07 [b] | 1.86 ± 0.11 [a] | 1.85 ± 0.02 [a] |
| Mo | 0.07 ± 0.01 [a] | 0.06 ± 0.01 [a] | 0.06 ± 0.01 [a] |
| Na | 80.38 ± 0.48 [c] | 197.03 ± 7.01 [a] | 151.35 ± 1.34 [b] |
| Ni | <LOQ | <LOQ | <LOQ |
| P | 746.83 ± 0.87 [c] | 1064.08 ± 6.44 [a] | 816.97 ± 8.81 [b] |
| Pb | <LOQ | <LOQ | <LOQ |
| S | 291.23 ± 2.17 [c] | 349.90 ± 2.02 [a] | 307.90 ± 1.02 [b] |
| Se | 0.02 ± 0.01 [a] | 0.02 ± 0.01 [a] | 0.01 ± 0.01 [a] |
| Zn | 8.17 ± 0.02 [c] | 9.79 ± 0.03 [a] | 9.03 ± 0.05 [b] |

Mean values with ±SD values and statistical analysis using Tukey's test at the significance level of $\alpha = 0.05$; the results indicated with the same letters do not demonstrate a statistically significant difference (test performed as part of a comparison of the concentrations of individual ions with each other after the application of slowly changing magnetic fields and with a control group); <LOQ—result below the Limit of Quantification.

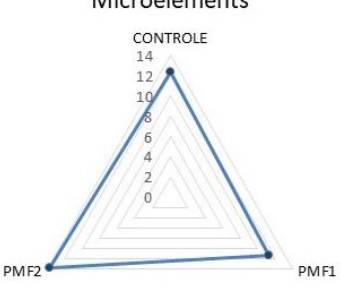

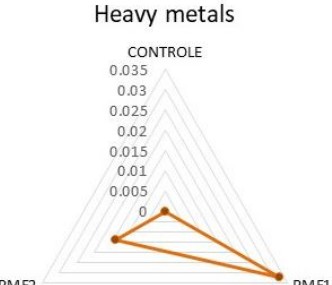

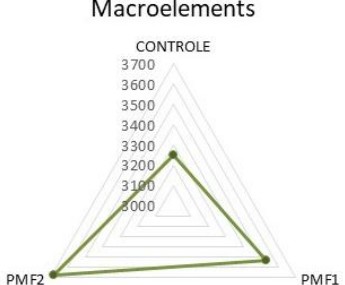

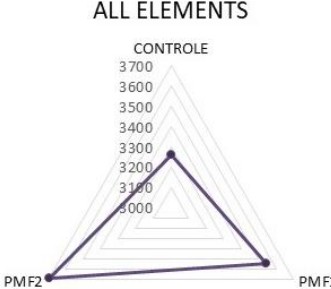

**Figure 6.** Changes in micro–macro elements and heavy metals and the sum thereof under the influence of the application of doses of pulsating magnetic field for cress sprouts.

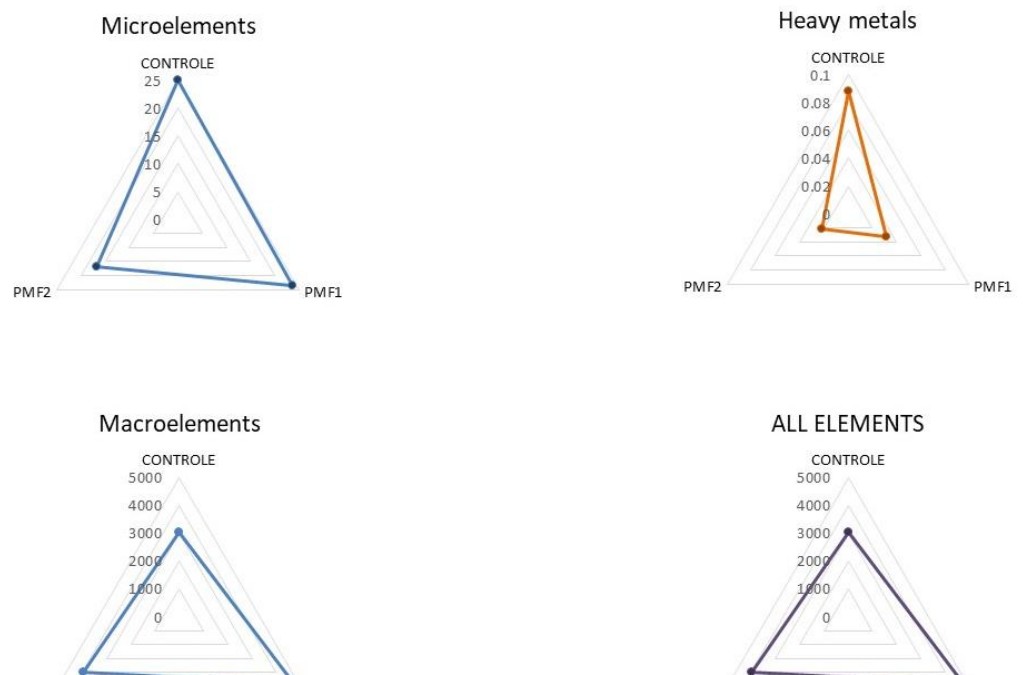

**Figure 7.** Changes in micro–macro elements and heavy metals and the sum thereof under the influence of the application of doses of pulsating magnetic field for radish sprouts.

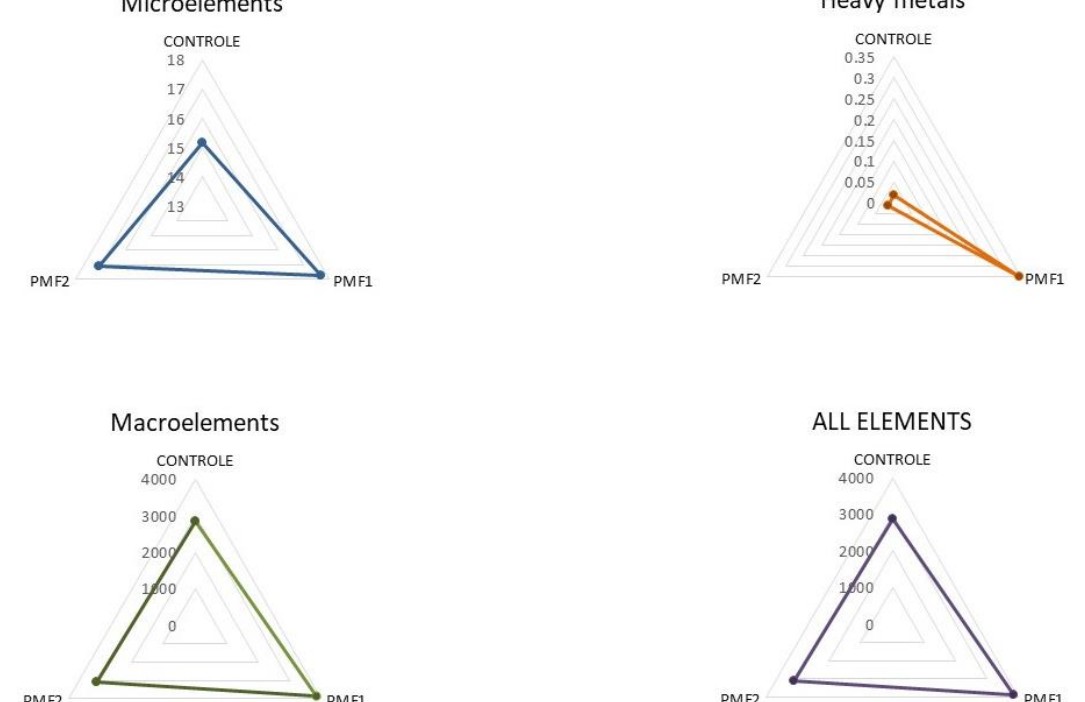

**Figure 8.** Changes in micro–macro elements and heavy metals and the sum thereof under the influence of the application of doses of the pulsating magnetic field for sunflower sprouts.

## 4. Conclusions

A magnetic field, especially a pulsating magnetic field, the essence of which is a fluctuating change in the direction of the magnetic field induction vector interaction over time, can be an excellent physical tool when used to accelerate seed germination. The authors proved that, in this case, a pulsating magnetic field with low induction reaching

100 μT and preferably frequencies of 100 Hz and 200 Hz, depending on the matrix of the subject of research, is particularly useful. The germination process starts much earlier when a pre-sowing pulsed magnetic field stimulant is applied for 30 min when compared to the control. This process allows for a significant improvement in the number of germinated seeds, especially by the 7–10th day of the seed germination process. The improvement is also noticeable in the quality of the sprouts, i.e., the content of basic macronutrients. This improvement depends on the species, with sunflower and radish sprouts storing the most nutritiously valuable macroelements under the influence of a pulsating field with a frequency of 100 Hz, while, for cress sprouts, it was with a pulsating field with a frequency of 200 Hz. These increases totalled 12% for cress sprouts, 57% for radish sprouts, and 38% for sunflower sprouts. These results are very significant, especially considering the nutritional value of plant sprouts. The above research should be continued and developed with new species of plant seeds and other parameters of the factor itself, i.e., the pulsating magnetic field.

**Author Contributions:** Conceptualization, G.Z.; formal analysis, G.Z., B.S.; investigation, G.Z., B.S., M.B.; resources, G.Z.; data curation, A.S., C.P.; writing—original draft preparation, G.Z.; writing—review and editing, G.Z., B.S., M.B., A.S.; visualization, A.S., B.S.; supervision, C.P.; project administration, B.S.; funding acquisition, C.P. All authors have read and agreed to the published version of the manuscript.

**Funding:** This research was funded by the Minister of Science and Higher Education in the range ofthe program entitled "Regional Initiative of Excellence" for the years 2019–2022, Project No.026/RID/2018/19, amount of funding 9 542 500.00 PL.

**Institutional Review Board Statement:** Not applicable.

**Informed Consent Statement:** Not applicable.

**Data Availability Statement:** Not applicable.

**Conflicts of Interest:** The authors declare no conflict of interest.

## Abbreviations

| | |
|---|---|
| **PMF** | pulsed magnetic field |
| **Al** | aluminum |
| **Ca** | calcium |
| **Cd** | cadmium |
| **Fe** | iron |
| **K** | potassium |
| **Mg** | magnesium |
| **Mn** | manganese |
| **Mo** | molybdenum |
| **Na** | sodium |
| **Ni** | nickel |
| **P** | phosphorus |
| **Pb** | lead |
| **S** | sulfur |
| **Se** | selenium |
| **Zn** | zinc |

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
