# Peer review of "Preliminary Research on the Influence of a Pulsed Magnetic Field on the Cationic Profile of Sunflower, Cress, and Radish Sprouts and on Their Germination Rate"

_applsci, doi:10.3390/app11209678_

Round 1
Reviewer 1 Report
• Please explain the reason for selecting these three crops for the study? • Arrange the sections properly; section third came before the second. • Provide the results from all the replications in the figures using error bars or box plots • Line 194: Why alfalfa is introduced here despite sunflower? Please provide information about alfalfa in the method section. • Work on the reference style in the manuscript, especially when you say X et al. have done this. • Arrange tables 2, 3, and 4, and Figures 3, 4, 5, 6, 7, and 8 in the same order. Follow the consistent pattern when you talk about these crops in the results section. • Provide the significance results for all the tests • The whole document need to be proofread for English editingAuthor Response
Dear Reviewer,
Thank You very much for Your kind suggestion.
We found these three plant sprouts very interesting due to their frequent use in our daily menu.
We have re-arranged the sections - they are now in the correct sequence and numbering.
We have rep[orted the results from all the replications in the figures using error bars.
We corected alfaalfa into sunflower - typographical error. We
We have revised our manuscript by a native speaker and in our reference style.
We corrected Figures and tables style and order.
We have reported the significance results for all the tests, except Fig 6-8 where summary results are presented in the form of radar charts where such an operation seems unnecessary.
Best Regards
Authors
Reviewer 2 Report
Well done work. A logical structure that is easy to follow. Tables and figures are clear.
I would like the authors to review the English language. Use a suffering sentence structure wherever possible. If this is done, I recommend publication.
Author Response
Dear Reviewer,
Thank You very much for Your kind suggestion. We have revised our manuscript by a native speaker.
Best Regards
Authors
Round 2
Reviewer 1 Report
Manuscript should be accepted for the publication